# Intensive Care as an Independent Risk Factor for Infection after Reconstruction and Augmentation with Autologous Bone Grafts in Craniomaxillofacial Surgery: A Retrospective Cohort Study

**DOI:** 10.3390/jcm10122560

**Published:** 2021-06-09

**Authors:** Jonas P. Jung, Kathrin Haunstein, Hans-Helge Müller, Ingo Fischer, Andreas Neff

**Affiliations:** 1Department of Oral and Maxillofacial Surgery, University of Marburg, Baldingerstrasse, D-35043 Marburg, Germany; kathrin.haunstein@web.de (K.H.); fischeri@med.uni-marburg.de (I.F.); andreas.neff@uk-gm.de (A.N.); 2Institute of Medical Bioinformatics and Biostatistics, University of Marburg, Bunsenstrasse 3, D-35032 Marburg, Germany; hans-helge.mueller@staff.uni-marburg.de

**Keywords:** ICU, risk factor, infection, reconstruction, augmentation, complication, prognostic model

## Abstract

Autologous bone grafts for reconstruction and augmentation are routinely used for maintaining functionality and facial aesthetics. Associated complications, however, have a significant impact on patients and health care systems. This study aims to investigate the possible risk factors associated with the occurrence of complications in order to provide evidence for the outcome of autologous bone graft reconstructive procedures. Patients from 2008 to 2018 who underwent autologous (mostly mandibular) reconstruction were included in the observational study. Clinical, pathological, and therapeutic factors were examined in univariate and multivariate analysis for significance with occurring complications. A multivariate model was used to create a prognostic model predicting the occurrence of complications. Graft complications requiring revision were exhibited by 33/128 patients. Infections were most frequent, with 4/22 patients affected by multi-resistant germs. Multivariate analysis showed radiotherapy (OR = 5.714; 95% CI: 1.839–17.752; *p* = 0.003), obstructive pulmonary disease (OPD) (OR = 4.329; 95% CI: 1.040–18.021; *p* = 0.044) and length of defect (in mm) (OR = 1.016; 95% CI: 1.004–1.028; *p* = 0.009) as independent risk factors associated with graft complications with high accuracy of prediction (AUC = 0.815). Intensive care (OR = 4.419; 95% CI: 1.576–12.388; *p* = 0.005) with a coefficient between intensive care and OPD (0.214) being low was identified as the most relevant risk factor for infection. Although intensive care is not a classic risk factor, but rather a summation of factors not reaching significance in the individual case, a stay in ICU (intensive care unit) needs to be considered for graft complications. As a clinical consequence, we recommend using the best possible hygienic measures during procedures e.g., while performing dressing and drainage changes in ICU.

## 1. Introduction

The integrity of the mandible, which is decisive for the form and functionality of the face and aesthetics, speech, chewing and swallowing process, is particularly at risk from tumor diseases, trauma, osteoradionecrosis, and chronic osteomyelitis. As the resulting loss of bone mass has a high impact on the patient’s quality of life, bone reconstruction and augmentation with fibula, iliac crest and scapula are common sites for autologous bone grafts, and therefore are of great importance in oral and maxillofacial surgery. The choice of graft depends on the anatomical location and size of the defect, with fibula grafts being the most used as the current “gold standard” [1,2]. The complication rate for reconstruction and augmentation procedures using autologous bone grafts in craniofacial surgery is reported with a greatly varying range between 10.3 to 69.2% [3,4,5,6,7,8,9,10,11,12,13,14,15,16,17,18], a re-operation rate of 10.6 to 58.2% [11,18,19,20,21] and a graft failure rate of 1.3 to 27% [3,9,10,11,13,20,21,22]. The correlation between graft failure and complication rate is controversially discussed with the type of graft used [2,20,23]. Age, defect size, radiotherapy and smoking are named as risk factors, with only radiotherapy and smoking recognized as independent risk factors [9,24]. Anasri et al. [3] describe a significant correlation between low skeletal muscle mass and the occurrence of complications in mandibular reconstructions using fibular grafts. Thus far, however, no reports have focused on mandibular reconstruction and intensive care. There is some experience-based information in the literature suggesting that there may be a relationship between complications in general and intensive care and the length of treatment in the intensive care unit (ICU). Alberti et al. [25] reported the incidence of infections in ICU to be as high as 21.1%. The incidence in patients with a long stay in ICU (>24 h) is higher at 32.3% than in patients with a short stay in ICU (<24 h) at 5.6%. The occurrence of an infection acquired in ICU in general has also been reported by Alberti et al. to be associated with the presence of a preexisting disease. The incidence in patients with at least one co-morbidity is 36.9%, and the incidence of patients without co-morbidity is 26.8%. Of the patients with a long stay (>24 h), 18.9% developed at least one infection acquired in the ICU. The incidence of ICU-acquired infections varied between 11.2% (95% CI: 9.2–13.1) in surgical ICU, 18.1% in internal ICU (95% CI: 16.4–19.8) and 20.7% in mixed ICUs (95% CI: 19.6–21.8) [25].

We therefore hypothesized that a stay on ICU might be an independent risk factor for complications, especially for infections after autologous reconstruction in OMFS. The aim of the present study is to examine whether there is a significant correlation between stays on the intensive care unit and other possible risk factors, such as comorbidities and underlying bone disease with the occurrence of complications after autologous bone grafts. We focused only on complications on the recipient side resulting in revision surgery after mandibular bone grafting.

## 2. Materials and Methods

The examined cohort was created using Operation and Procedure Codes (OPS) from the patient collective of the Department of Oral and Maxillofacial Surgery at the University of Marburg (Table 1). Inclusion criteria were surgical procedure between 2008 to 2018, reconstruction or augmentation with autologous bone grafts and fixation with osteosynthesis devices (i.e., miniplating and/or reconstruction plates). Exclusion criteria were reconstruction or augmentation with autogenous bone grafts and procedures without plate based osteosynthesis devices. Ethical approval was waived by the local Ethics Committee of University of Marburg in view of the retrospective nature of the study and as all the procedures performed were part of the routine care (file number: ek_mr_24_04_19_neff).

The following data were collected from the paper and digital records: (a) surgery-related data: type of graft (vascularized fibula graft versus non-vascularized iliac crest graft), length of defect (mm) and number of graft osteotomies, mandibular and maxillary defects classified according to Brown [26,27], operation duration (minutes), type of plate used (mini plates versus reconstruction plates), experience of the surgeon (resident versus consultant); (b) patient-related data: patient ID, age (in years), gender (male versus female), American Society of Anesthesiology (ASA) class ≥ 3 (yes versus no), intensive care after surgery (yes versus no), time on ICU (in days); comorbidities: hypertension (yes versus no), smokers (yes versus no), obstructive pulmonary disease (OPD) (yes versus no), diabetes mellitus (yes versus no), regular alcohol use (yes versus no), radiotherapy (yes versus no); (c) complications observed: complication (yes versus no), type of complication (Table 2), time from surgery to re-operation (in days), identified problematic and/or multi-resistant germs (e.g., MRSA and Pseudomonas aeruginosa) (yes versus no). Complications were further differentiated between early complications (revision surgery performed within 30 days) and late complications (revision surgery performed after 30 days), however, both groups were combined for statistical analysis, because only one patient underwent revision surgery within 30 days. The complications only refer to the recipient side of the graft, donor site morbidity was not assessed.

To investigate the factor “intensive care” more in detail, factors specific to intensive care medicine were collected from patients on ICU: SpO2 (in %), FiO2 (in %), pO2 (in mmHg), pCO2 (in mmHg), pH, mechanical ventilation (yes versus no) and Hb (in g/dL).

## 3. Statistical Analysis

Continuous variables are shown with mean ± standard deviation, categorical variables with absolute and relative frequency. A univariate analysis was performed with the *t*-test for continuous normally distributed variables, the Mann–Whitney test for continuous non-normally distributed variables. The chi^2^ test was used at categorical variables and the exact Fisher test for expected frequencies *n* < 5. The statistically significant variables from the univariate analysis were examined for significance in the multivariate logistic regression. Potential risk factors associated with complications in general, infections in general and infections with problematic germs were investigated.

A *p*-value of less than 0.05 is statistically significant and the strength of the association between two events is shown by odds ratio. Note that a multiplicative factor in the form of an OR always refers to a unit (length of defect in mm, operation duration in minutes, age in years, ICU in days, time from surgery to re-operation in days. Data were collected using Excel 2016 (Microsoft Corporation, Redmond, WA, USA) and analyzed using SPSS Statistics 20 (IBM Corporation, Armonk, NY, USA).

## 4. Results

Based on the inclusion criteria, 128 patients were detected in the cohort with autologous reconstruction of the mandible (*n* = 108/128, 84.4%), maxilla (*n* = 13/128, 10.2%), mandible and maxilla (*n* = 2/128, 1.6%), and other regions of the facial skeleton (*n* = 5/128, 3.9%), i.e., temporomandibular fossa (*n* = 2), zygoma (*n* = 1) nasal bone and midface (*n* = 2). In 38 (29.9%) cases, vascularized fibula grafts were implanted, in 89 (70.1%) cases non-vascularized iliac crest grafts. Mean operation duration was 368 ± 220 min and the length of defect was 75 ± 47 mm. The mandibular and maxillary defects are classified according to Brown [26,27] (Table 3). In 79 (80.6%) cases a reconstruction was performed using mini plates for fixation, in 19 (19.4%) cases reconstruction plates were used.

Of the overall collective, 74 (57.8%) of patients were male and 54 (42.2%) of patients were female. Mean age of the cohort was 52.7 ± 17.6 years. After surgery, 43 (33.6%) patients stayed in ICU and spent a mean of 4.86 ± 4 days in ICU. During surgery 75 (65.8%) patients were classified in ASA classes less than 3 and 39 (34.2%) in ASA classes 3 or above. Regarding comorbidities, 36 (30.3%) of the patients exhibited hypertension, 47 (39.8%) were smokers, 14 (11.8%) had an obstructive pulmonary disease (OPD/asthma), 13 (10.9%) diabetes mellitus and 29 (24.6%) drank alcohol regularly. Out of all patients, 27/128 (21.1%) received pre- or postoperative radiotherapy. An overview of the data collected is given in Table 4.

A total of 33 patients with 59 complications were included in this study. There were 16 patients (12.5% of the collective and 48% of the patients with complications, respectively) with 1 complication, 9 patients (7% or 27.3%, respectively) with 2 complications, 7 patients (5.5% or 21.2%, respectively) with 3 complications and 1 patient (0.8% or 3%, respectively) with 4 complications. There was an overall complication rate of 25.8% and a graft loss rate of 3.1%. The evaluation refers to the recipient side of the graft. Ranking from most to least frequent, complications were infection (*n* = 22/59 (37.3%)) of all complications/*n* = 22/128 (17.2%) of all patients, respectively), non-union (*n* = 15/59 (25.4%)/*n* = 15/128 (1.7%)), wound dehiscence (*n* = 5/59 (8.5%)/*n* = 5/128 (3.9%)), necrosis (*n* = 4/59 (6.8%)/*n* = 4/128 (3.1%)), graft failure (*n* = 4/59 (6.8%)/*n* = 4/128 (3.1%)), plate exposure (*n* = 4/59 (6.8%)/*n* = 4/128 (3.1%)), loose plate (*n* = 2/59 (3.4%)/(*n* = 2/128 (1.6%)), plate fracture (*n* = 1/59 (1.7%)/*n* = 1/128 (0.8%)), isolated bone exposure (*n* = 1/59 (1.7%)/*n* = 1/128 (0.8%)) and fistula (*n* = 1/59 (1.7%)/*n* = 1/128 (0.8%)) (Table 2). From the 22 Patients with infection, 4 (4/22, 18.2%) had an infection with problematic germs.

Our statistical evaluation can be divided into three parts examining possible risk factors with the occurrence of complications, (a) in general, (b) with infection as the most frequent occurring complication, and (c) infections with problematic germs. Possible risk factors were assessed for significance using univariate and multivariate analysis.

The univariate analysis of possible risk factors with complications in general shows significant correlations with gender, age, intensive care, ASA class ≥ 3, radiotherapy, hypertension, OPD, type of graft, length of defect, operation duration and number of graft osteotomies. An overview of the univariate analysis of the factors for complications is presented in Table 5. According to the multivariate analysis radiotherapy (OR = 5.714; 95% CI: 1.839–17.752; *p* = 0.003), OPD (OR = 4.329; 95% CI: 1.040–18.021; *p* = 0.044) and length of defect in mm (OR = 1.016; 95% CI: 1.004–1.028; *p* = 0.009) are independent risk factors for the occurrence of complications (Table 6). The coefficient between the risk factors radiotherapy, OPD and length of defect can be described as low at 0.149 and moderate at 0.323 and 0.326 (Table 7). Gender, age, intensive care, days on ICU, ASA class ≥ 3, hypertension, type of graft, operation duration and number of graft osteotomies are removed by forward elimination from the logistic regression and are not listed in our model. In total, the model achieves significance in the Omnibus-Test (*p* < 0.001) and has a Nagelkerke-R^2^ outcome of R^2^ = 0.376. The results have an effect size of f = 0.78 which counts as a large effect size according to Cohen (1988). Based on these factors, this logistic regression model has the biggest influence in our study. Using the constant and the odds from the model of multivariate analysis, the prediction score *p* of the 33 complication patients were calculated (Figure 1). The occurrence of complications in the patient collective is shown under dependence on the prediction score. A linear relationship between the prediction score and the probability of complications can be seen (Figure 2). This diagnostic predictive model is shown by sensitivity and specificity in the Receiver Operating Characteristic (ROC) curve (Figure 3). The area under the curve (AUC) is 0.815.

The univariate analysis of possible risk factors with infection shows a significant correlation with gender, age, intensive care, ASA class ≥ 3, radiotherapy, hypertension, obstructive pulmonary disease, as well as type of graft, length of defect, operation duration and number of graft osteotomies (Table 8). The multivariate analysis shows that intensive care (OR = 4.419; 95% CI: 1.576–12.388; *p* = 0.005) and OPD (OR = 4.388; 95% CI: 1.252–15.375; *p* = 0.021) are independent risk factors for the occurrence of infection (Table 9). The coefficient between the risk factors intensive care and OPD can be described as low at 0.214.

The univariate analysis of possible risk factors with infection with problematic/multi-resistant germs shows significant correlation with intensive care, radiotherapy, and length of defect. (Table 10). Since the events were too small (*n* = 4), no multivariate analysis was performed.

The specific intensive care factors: saturation of peripheral oxygen (SpO_2_), fraction of inspired oxygen (FiO_2_), oxygen partial pressure (pO_2_), carbon dioxide partial pressure (pCO_2_), potential of hydrogen (pH), mechanical ventilation and hemoglobin (Hb) from patients on ICU did not show a significant association for complications in general or from infections and infections with problematic germs (Table 11, Table 12 and Table 13).

Regarding complications, the logistic regression of the possible risk factors shows that radiotherapy, OPD and the length of defect are significant factors associated with postoperative complications such as wound infections with problematic/multi-resistant germs (37.5%) followed by non-union (25.4%), i.e., those with wound infection ranking by far most frequent among the complications observed. The variables mentioned above are considered as independent risk factors. The other possible risk factors such as gender, age, ASA class ≥ 3, hypertension and type of graft show a significant correlation in the univariate analysis. Nevertheless, these are removed from our model in the logistic regression as cofounders and, therefore, cannot be accepted as independent risk factors in our study. The underlying rationale is that the elimination in logistic regression is based on testing all variables whether their contribution is significant after adding new variables. This can lead to the elimination of an already selected variable if this variable has become redundant because of its relationship with the other variables. Thus, the automatic procedure leads to the selection of predictive variables. Factors can initially reach significance in univariate analysis but not in multivariate analysis. For this reason, we do not consider these parameters as independent risk factors.

## 5. Discussion

The reconstruction of the jaw and the bones of the facial skeleton represents a major challenge. Complications and revision surgeries are serious burdens for the patient and for the health care system [28,29]. However, reconstruction with autologous bone grafts is the current gold standard and allows for the greatest gain in quality of life [29,30,31]. With this study, possible risk factors could be identified showing a significant correlation with the occurrence of complications in general, and especially with infections and infections with problematic germs. The complication rate of 25.8% in the cohort is in the lower to medium range of the 10.3 to 69.2% reported as having a high variability by comparable studies [3,4,5,6,7,8,9,10,11,12,13,14,15,16,17,18]. Regarding our complication rate, the fact should be considered that we have only recorded complications that required a surgical revision procedure after bone reconstruction and augmentation on the recipient side, and that we did not assess donor site morbidity including wound healing complications such as those observed rather frequently after skin-split grafts at the donor site. Compared to other studies, the graft loss rate of 3.1% is in the lower range of 1.3 to 27% despite a high percentage of free iliac crest grafts in the present study [3,9,10,11,13,20,21,22].

As noted above, according to our logistic regression radiotherapy, OPD and the length of defect were detected as significant factors associated with postoperative complications such as wound infections with problematic/multi-resistant germs (37.5%) followed by non-union (25.4%), with wound infection ranking by far most frequent among the complications observed. This was contrary to our expectation, as—from a clinical point of view—the type of graft used (i.e., in our group, microvascularized fibula flaps versus free iliac grafts) and the kind of osteosynthesis used are usually considered to rank among the most relevant factors for potential complications such as non-union and plate fractures, etc. [32]. According to our findings, at least from a statistical standpoint, these factors did not provide evidence to be decisive for complication rates in general.

The fibula graft is currently the most commonly used graft for mandibular reconstruction and allows for multiple segmentation, i.e., multiple osteotomies for optimal shape adaptation [9,33,34]. According to a systematic review (9499 defects), the overall rate of graft failure is 4.1% for the fibula and is highest for the iliac crest with 6.2% [2], although Lonie et al. [23] contradict this in their meta-analysis by not seeing any relevant differences between the iliac crest and fibula. This statement would be in line with the statistical results of our study, with a rate of graft failures at 3.1%, and moreover the rate of graft failure for the iliac crest at 3.4% is higher than the rate of graft failure of the fibula at 2.6%. Nevertheless, there is no statistical significance in the multivariate analysis between the types of graft and the occurrence of complications in general, infections, or infections with problematic germs in our study. Other potential complications described in literature include plate fracture, fistula formation, bone necrosis, and non-union [35,36]. In the literature, regardless of the type of osteosynthesis used (mini plates versus reconstruction plates), a rate of non-union of up to 14% and 13%, respectively, has been reported for segmental defects in fibular grafts. Although plate fractures were significantly more common in mini-plate osteosynthesis (10% vs. 0%) [9,24], the problem is primarily due to the compromised hard and soft tissues following adjuvant therapies (radiotherapy, radio-chemotherapy). The rate of non-union is also reported to depend significantly on the type of graft used. For microvascular reconstructions, for which the rate of non-union was still given to be as high as 7–9% in the older literature [2,9], however according to current review literature, the rate is reported with about 5%, although it may be supposed that there is a “surprisingly high reporting bias, especially since the associated morbidity is relevant” [2]. In our study, the rate of non-union in the microvascular fibula grafts was also higher at 26.3% (10/38) than in the non-microvascular iliac crest at 4.5% (4/89). We explain the high rate of non-union in fibula grafts by the small size of the patient group. According to the review, osteocutaneous radial flaps and scapular flaps show significantly higher non-union rates of 9.1% (78/866) and 13.1% (49/375), respectively, compared to fibular grafts with 3.9% (103/2632). Tight non-union without need for clinical intervention is reported for scapular grafts even in up to 39% of reconstructions. The iliac crest grafts perform best in this respect with 2.6% (17/646) [2].

Like Liu et al. [7] and Chen et al. [24] we could also verify radiotherapy as an independent risk factor for complications after jaw reconstruction and augmentation with autologous bone grafts (OR = 5.714, 95% CI: 1.839–17.752, *p* = 0.003), being responsible for compromised hard and soft tissues following adjuvant therapies (radiotherapy, radio-chemotherapy). In our study, against our expectation, smoking showed no significant correlation with the occurrence of complications (*p* = 0.437). This could be explained by the large smoking proportion of patients (39.8%) in the cohort, resulting in a balanced distribution in both groups.

Obstructive pulmonary disease was mentioned by Hanasono et al. [16] as a possible risk factor for the occurrence of complications in jaw reconstruction but did not attain significance in the univariate and in the multivariate analysis. Likewise, obstructive pulmonary disease has not been listed in one of the comparable studies as a significant risk factor for the occurrence of complications after reconstruction and augmentation with autologous bone grafts but may explain a prolonged stay on the ICU. Thus, obstructive pulmonary disease is likely to heighten the risk of nosocomial infections, which, in turn, may explain the high rate of postoperative infections (22/59) among the complications observed in our collective (i.e., 37.3%). This effect is confirmed by Häfner et al. [37], who found that patients in ICU have a 5 to 10 times higher risk of infection than patients in normal ward. In their study, a total of 36,999 patients were included, of which 3.5% suffered from nosocomial infection. Leading infections when considering the total patient population were postoperative wound infection with 24.7%, which is quite in line with the overall wound infection rate of 25.8% (22/128) among all reconstructions/augmentations in our study.

In this study the calculated incidence of infections on ICU is 34.1%. All ICU patients spent more than 24 h on ICU. Alberti et al. [25] specified the incidence of infections of patients with long stay on ICU (>24 h) with 32.3%. Compared to our study, both incidences are similar. To the best of our knowledge, so far, no comparison of intensive care (viz. postoperative stay on the ICU) and the occurrence of complications, especially infections after reconstruction of the facial skeleton with autologous bone grafts (as presented in this study) has been published, and no comparable diagnostic tests for the occurrence of complications after reconstruction and augmentation with autologous bone grafts in craniomaxillofacial surgery have been described in any of the studies we could retrieve through our literature search. To explain the strong correlation between intensive care and the occurrence of infection and infection with problematic germs, additional parameters were collected on ICU, which we suspected could be responsible for this correlation (Table 11, Table 12 and Table 13). However, none of these parameters showed a significant correlation with the occurrence of complications in the univariate analysis. Consequently, it is not possible to explain the strong correlation between intensive care and the occurrence of infection and infection with problematic germs by individual factors.

One of the major limitations of our study is the heterogeneity and the small size of the patient group. With only 33 patients with complications, only 3 characteristics can be indicated as independent factors in the multivariate model. To create stable multivariate models with several factors, larger numbers of patients are needed. In particular, a larger number of events is needed for a more accurate analysis of the risk factors for infections with problem germs. Although we did our best to assess a wide range of surgery and patient-related data, an investigation of such a complex problem, due to its retrospective design, is limited by depending on data collected via daily routine work. A variety of individual factors related to both surgical procedure and patients’ individual predisposition, therefore, may go unnoticed. We are aware that radiotherapy, OPD and length of defect are just exemplary drivers for the occurrence of complications and that intensive care is not the only factor responsible for the occurrence of infections. Nevertheless, these factors are results with statistical significance from the univariate and multivariate analysis and could be shown to be independent risk factors. The purpose of this article is to draw attention to this yet unknown correlation, especially regarding the correlation between infections and stay on ICU.

Our study results, therefore, may help to estimate the probability of the occurrence of a complication after reconstruction and augmentation with autologous bone grafts using the risk factors radiotherapy, OPD and length of defect. The accuracy of the diagnostic prediction is good with an AUC = 0.815.

In line with our initial hypothesis, intensive care regression in fact showed the strongest effect on infection as an independent risk factor with an OR of 4.419 in the multivariate logistic, besides radiotherapy (OR 5.714), for complications in general. In the univariate analysis, intensive care showed the highest association with the occurrence of infections with problematic germs with an OR of 19.48. An independent correlation could not be shown because of the small number of events (*n* = 4), but is suspected. Thus, intensive care has a greater impact on the incidence of infection than other patient-related factors such as age, underlying diseases, multimorbidity, size of the defect and the surgical complexity. All of these factors show a less significant influence on the association with both infection and infection with problematic germs in multivariate analysis and are subordinate to intensive care as an independent risk factor in our model. Our results therefore indicate that patients with an ICU stay are more likely to develop a postoperative infection than patients with a long list of comorbidities treated in non-ICU wards.

It should be noted that it is not the intention of the present study to describe intensive care as something detrimental. The ICU is an important and indispensable component for an optimal and appropriate treatment of patients. More importantly, we would like to draw attention to the significant correlation discussed in this study. We see intensive care not as a classic risk factor, but more than a summation of unknown factors and known factors, which do not reach significance in the individual case.

## 6. Conclusions

As graft infections are associated, first of all, with stays on the ICU, especially in risk patients (i.e., with obstructive pulmonary disease), our therapeutic recommendation is the best possible intensification of hygienic measures during procedures such as dressing and redon drainage changes in ICU. As a consequence, we now avoid, for example, penrose or easyflow drainages for microvascular anastomoses. If infections occur after intensive medical ICU treatment, we recommend a fast determination of germ resistance and a corresponding adjustment of the antibiotic treatment. Whether these measures reduce the occurrence of complications must be investigated in the future.

## Figures and Tables

**Figure 1 jcm-10-02560-f001:**
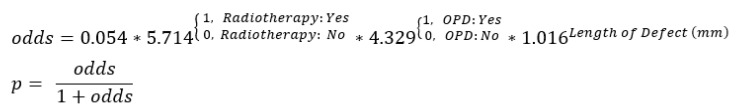
The Calculation of the Prediction Score *p*.

**Figure 2 jcm-10-02560-f002:**
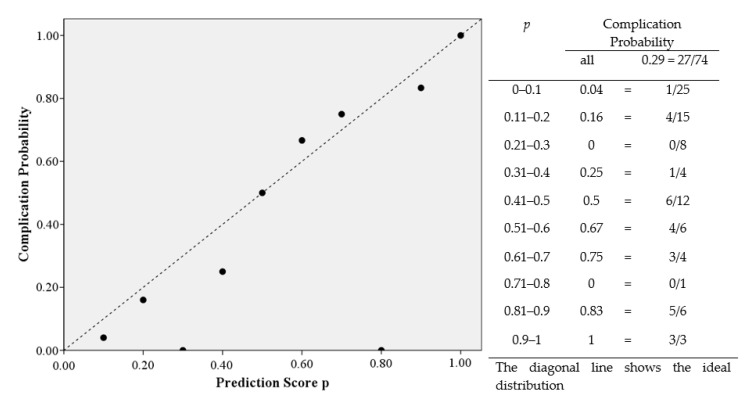
Complication Probability depending on the Prediction Score *p*.

**Figure 3 jcm-10-02560-f003:**
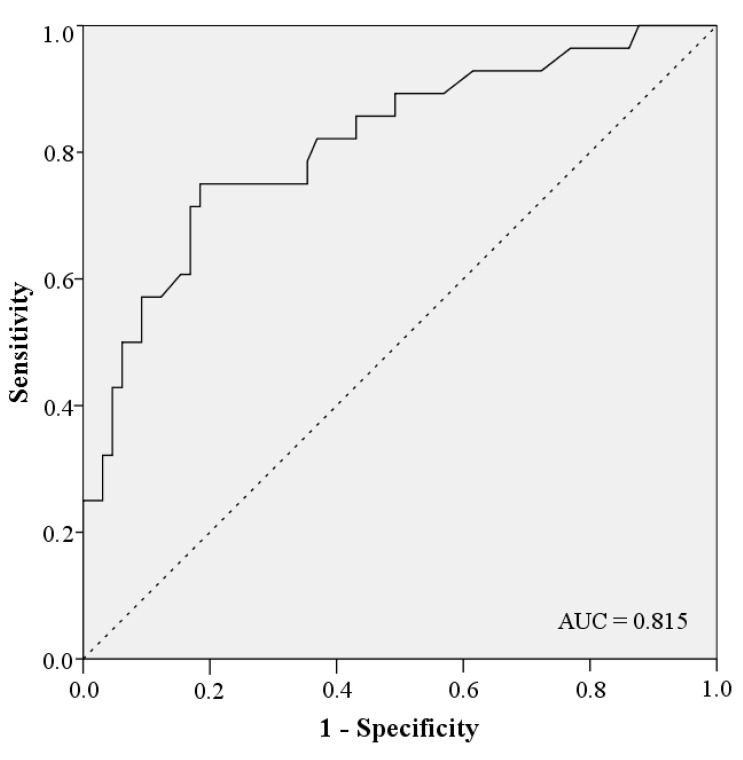
ROC Curve.

**Table 1 jcm-10-02560-t001:** Operation and procedure codes.

OPS	Description
5-771	Partial and total resection of a facial skull bone
5-772	Partial and total resection of the mandible
5-77b	Bone transplantation and transposition on jaw and facial skull bones
5-783	Harvesting of a bone graft
5-784	Bone transplantation and transposition

**Table 2 jcm-10-02560-t002:** Overview of the Type of Complications Observed.

Complications at Recipient Site	*n*
Infection	22
Non-union	15
Wound dehiscence	5
Necrosis	4
Graft failure	4
Plate exposure	4
Loose plate	2
Plate fracture	1
Isolated bone exposure	1
Fistula	1

Numbers in absolute frequency.

**Table 3 jcm-10-02560-t003:** Distribution of Mandibular and Maxilla Defects of the Patient Cohort According to the Brown Classification System.

Classification of the Mandibular and Maxilla Defect According to Brown.		
Mandibular defects [26]	Class I	51
	Class Ic	2
	Class II	12
	Class IIc	2
	Class III	10
	Class IV	11
	Class IVc	1
Maxillary defects [27]	Class I	6
	Class II	2
	Class III	1
	Class IV	0
	Class V	0
	Class VI	3

Numbers in absolute frequency.

**Table 4 jcm-10-02560-t004:** Overview of the Data Collected from the Patient Cohort.

Overview of the Data Collected		*n* (%)
**(a) ** **Surgery-Related Data**		
Type of graft	Vascular fibula	38 (29.9)
	Non-vascular iliac crest	89 (70.1)
Length of defect in mm (mean ± SD)		75 ± 47

Number of graft osteotomies	0	85 (72.6)
	1	16 (13.7)
	2	16 (13.7)
Operation duration in minutes (mean ± SD)		368 ± 220

Plates	Reconstruction plates	19 (19.4)
	Mini plates	79 (80.6)
Surgeon experience	Resident	16 (13.3)
	Consultant	104 (86.7)
**(b) ** **Patient-Related Data**		
Age in years (mean ± SD)		52.7 ± 17.6

Gender	Male	74 (57.8)
	Female	54 (42.2)
ASA Class ≥ 3	Yes	39 (34.2)
	No	75 (65.8)
Intensive care	Yes	43 (33.6)
	No	85 (66.4)
ICU in days (mean ± SD)		4.86 ± 4

Hypertension	Yes	36 (30.3)
	No	83 (69.7)
Smoker	Yes	47 (39.8)
	No	71 (60.2)
OPD	Yes	14 (11.8)
	No	105 (88.2)
Diabetes mellitus	Yes	13 (10.9)
	No	106 (89.1)
Regular alcohol use	Yes	29 (24.6)
	No	89 (75.4)
Radiotherapy	Yes	27 (22.5)
	No	93 (77.5)
**(c) ** **Complications Observed**		
Complications	Yes	33 (25.8)
	No	95 (74.2)
Time from surgery to re-operation in days		165.3 ± 167.7
Infection with problematic Germs	Yes	4 (3.1)
	No	124 (96.9)

**Table 5 jcm-10-02560-t005:** Univariate Analysis of Risk Factors on Complication.

		Complication		*p*	OR
		Yes, *n* (%)	No, *n* (%)		
**(a) ** **Surgery-Related Data**					
Type of graft	Vascular fibula	18 (14.2)	20 (15.8)	**<0.001**	4.440 [1.908; 10.33]
	Non-vascular iliac crest	15 (11.8)	74 (58.3)		
Length of defect (mean ± SD)		103 ± 53.6	63 ± 38.1	**<0.001**	1.019 [1.008; 1.031]

Number of graft osteotomies	0	11 (9.4)	74 (63.2)	**<0.001**	3.347 [1.877; 5.967]
	1	9 (7.7)	7 (6.0)		
	2	9 (7.7)	7 (6.0)		
Operation duration (mean ± SD)		470 ± 212	332 ± 211	**0.004**	1.003 [1.001; 1.005]
Plates	Reconstruction plates	9 (9.2)	10 (10.2)	0.078	2.486 [0.888; 6.960]
	Mini plates	21 (21.4)	58 (59.2)		
Surgeon experience	Resident	2 (1.7)	14 (11.7)	0.231	0.352 [0.076; 1.646]
	Consultant	30 (25.0)	74 (61.7)		
**(b) ** **Patient-Related Data**					
Age (mean ± SD)		58.9 ± 14.7	50.6 ± 18.0	**0.035**	1.029 [1.004; 1.056]

Gender	Male	25 (19.5)	49 (38.3)	**0.015**	2.934 [1.202; 7.159]
	Female	8 (6.3)	46 (35.9)		
ASA Class ≥ 3	Yes	17 (14.9)	22 (19.3)	**0.008**	3.091 [1.323; 7.224]
	No	15 (13.2)	60 (52.6)		
Intensive care	Yes	22 (17.2)	21 (16.4)	**<0.001**	7.048 [2.950; 16.840]
	No	11 (8.6)	74 (57.8)		
ICU in days (mean ± SD)		3.3 ± 4.5	1.1 ± 2.5	**<0.001**	1.221 [1.062; 1.404]
Hypertension	Yes	16 (13.5)	20 (16.8)	**0.007**	3.106 [1.332; 7.240]
	No	17 (14.3)	66 (55.5)		
Smoker	Yes	15 (12.7)	32 (27.1)	0.437	1.380 [0.612; 3.114]
	No	18 (15.3)	53 (44.9)		
OPD	Yes	9 (7.6)	5 (4.2)	**0.003**	6.075 [1.859; 19.856]
	No	24 (20.2)	81 (68.1)		
Diabetes mellitus	Yes	6 (5.0)	7 (5.9)	0.185	2.508 [0.775, 8.119]
	No	27 (22.7)	79 (66.4)		
Regular alcohol use	Yes	9 (7.6)	20 (17.0)	0.672	1.219 [0.488; 3.044]
	No	24 (20.3)	65 (55.1)		
Radiotherapy	Yes	16 (13.3)	11 (9.2)	**<0.001**	6.503 [2.564; 16.490]
	No	17 (14.2)	76 (63.3)		

**Table 6 jcm-10-02560-t006:** Multivariate Analysis of Significant Risk Factors on Complication in the Univariate Analysis.

	OR	95% CI	*p*
Radiotherapy	5.714	1.839–17.752	0.003
OPD	4.329	1.040–18.021	0.044
Length of defect	1.016	1.004–1.028	0.009

**Table 7 jcm-10-02560-t007:** Correlation between Radiotherapy, OPD and Length of Defect.

	Radiotherapy	OPD	Length of Defect
Radiotherapy	1	0.323	0.326
OPD		1	0.149
Length of defect			1

**Table 8 jcm-10-02560-t008:** Univariate Analysis of Risk Factors on Infection.

		Infection		*p*	OR
		Yes, *n* (%)	No, *n* (%)		
**(a) ** **Surgery-Related Data**					
Type of graft	Vascular fibula	11 (8.7)	27 (21.3)	**0.024**	2.889 [1.125; 7.421]
	Non-vascular iliac crest	11(8.7)	78 (61.4)		
Length of defect (mean ± SD)		109.3 ± 54.6	66.8 ± 41.0	**0.001**	1.018 [1.006; 1.029]

Number of graft osteotomies	0	7 (6.0)	78 (66.7)	**0.001**	2.765 [1.511; 5.060]
	1	6 (5.1)	10 (8.5)		
	2	6 (5.1)	10 (8.5)		
Operation duration (mean ± SD)		455.8 ± 219.9	350.8 ± 215.3	0.071	1.002 [0.9998; 1.004]

Plates	Reconstruction plates	9 (9.2)	10 (10.2)	**0.003**	5.564 [1.846; 16.765]
	Mini plates	11 (11.2)	68 (69.4)		
Surgeon experience	Resident	2 (1.7)	14 (11.7)	0.439	1.565 [0.328; 7.468]
	Consultant	19 (15.8)	85 (70.8)		
**(b) ** **Patient-Related Data**					
Age (mean ± SD)		60.7 ± 12.4	51.1 ± 18.1	**0.022**	1.036 [1.005; 1.068]

Gender	Male	17 (13.3)	57 (44.5)	**0.034**	2.922 [1.004; 8.502]
	Female	5 (3.9)	49 (38.3)		
ASA Class ≥ 3	Yes	11 (9.6)	64 (56.1)	0.082	5.235 [0.813; 8.235]
	No	11 (9.6)	28 (24.6)		
Intensive care	Yes	15 (11.7)	28 (21.9)	**<0.001**	5.969 [2.206; 16.156]
	No	7 (5.5)	78 (60.9)		
ICU in days (mean ± SD)		3.4 ± 4.9	1.3 ± 2.7	**0.001**	1.162 [1.025; 1.317]
Hypertension	Yes	12 (10.1)	24 (20.2)	**0.006**	3.650 [1.401; 9.501]
	No	10 (8.4)	73 (61.3)		
Smoker	Yes	9 (7.6)	38 (32.2)	0.909	1.057 [0.411; 2.714]
	No	13 (11.0)	58 (49.2)		
OPD	Yes	7 (5.9)	7 (5.9)	**0.004**	6.000 [1.841; 19.559]
	No	15 (12.6)	90 (75.6)		
Diabetes mellitus	Yes	5 (4.2)	8 (6.7)	0.063	3.272 [0.955; 11.216]
	No	17 (14.3)	89 (74.8)		
Regular alcohol use	Yes	6 (5.1)	23 (19.5)	0.745	1.190 [0.417; 3.397]
	No	16 (13.6)	73 (61.9)		
Radiotherapy	Yes	11 (9.2)	16 (13.3)	**0.002**	5.125 [1.899; 13.830]
	No	11 (9.2)	82 (68.3)		

**Table 9 jcm-10-02560-t009:** Multivariate Analysis of Significant Risk Factors on Infection in the Univariate Analysis.

	OR	95% CI	*p*
Intensive Care	4.419	1.576–12.388	0.005
OPD	4.388	1.252–15.375	0.021

**Table 10 jcm-10-02560-t010:** Univariate Analysis of Risk Factors on Infection with problematic germs.

		Infection with Problematic Germs		*p*	OR
		Yes, *n* (%)	No, *n* (%)		
**(a) ** **Surgery-Related Data**					
Type of graft	Vascular fibula	2 (1.6)	36 (28.3)	0.346	2.417 [0.328; 17.823]
	Non-vascular iliac crest	2 (1.6)	87 (68.5)		
Length of defect (mean ± SD)		142.3 ± 75.5	71.7 ± 42.9	**0.047**	1.021 [1.004; 1.039]

Number of graft osteotomies	0	1 (0.9)	84 (71.8)	0.058	2.409 [0.788; 7.371]
	1	2 (1.7)	14 (12.0)		
	2	1 (0.9)	15 (12.8)		
Operation duration (mean ± SD)		405.3 ± 179.8	366.7 ± 220.9	0.645	1.001 [0.996; 1.005]

Plates	Reconstruction plates	2 (2.0)	17 (17.3)	0.168	4.529 [0.595; 34.458]
	Mini plates	2 (2.0)	77 (78.6)		
Surgeon experience	Resident	0 (0.0)	16 (13.3)	0.560	0.676 [0.035; 13.163]
	Consultant	4 (3.3)	100 (83.3)		
**(b) ** **Patient-Related Data**					
Age (mean ± SD)		61.8 ± 6.9	52.5 ± 17.8	0.261	1.035 [0.969; 1.106]

Gender	Male	3 (2.3)	71 (55.5)	0.436	0.447 [0.045; 4.414]
	Female	1 (0.8)	53 (41.4)		
ASA Class ≥3	Yes	1 (0.8)	74 (61.7)	0.115	0.162 [0.016; 1.614]
	No	3 (2.5)	36 (30.0)		
Intensive care	Yes	4 (3.1)	39 (30.5)	**0.012**	19.481 [1.024; 370.728]
	No	0 (0.0)	85 (66.4)		
ICU in days (mean ± SD)		2.3 ± 0.4	1.6 ± 3.3	0.061	1.045 [0.823; 1.328]
Hypertension	Yes	3 (2.5)	33 (27.7)	0.082	7.455 [0.748; 74.279]
	No	1 (0.8)	82 (68.9)		
Smoker	Yes	2 (1.7)	45 (38.1)	0.523	1.533 [0.208; 11.281]
	No	2 (1.7)	69 (58.5)		
OPD	Yes	2 (1.7)	12 (10.1)	0.068	8.583 [1.106; 66.618]
	No	2 (1.7)	103 (86.6)		
Diabetes mellitus	Yes	1 (0.8)	12 (10.1)	0.374	2.861 [0.275; 29.727]
	No	3 (2.5)	103 (86.6)		
Regular alcohol use	Yes	1 (0.8)	28 (23.5)	0.682	1.024 [0.102; 10.243]
	No	3 (2.5)	86 (72.3)		
Radiotherapy	Yes	3 (2.5)	24 (20.0)	**0.035**	11.500 [1.144; 115.552]
	No	1 (0.8)	92 (76.7)		

**Table 11 jcm-10-02560-t011:** Univariate Analysis of Variables from ICU Patients on Complications.

		Complications		*p*
		Yes (*n* = 22)	No (*n* = 21)	
SpO_2_ (%) (mean ± SD)		97.8 ± 2.7	98.2 ± 1.3	0.656
FiO_2_ (%) (mean ± SD)		57.6 ± 23.0	54.7 ± 20.2	0.688
pO_2_ (mmHg) (mean ± SD)		155.6 ± 86.5	148.5 ± 69.8	0.780
pCO_2_ (mmHg) (mean ± SD)		40.4 ± 7.3	40.2 ± 5.0	0.652
pH (mean ± SD)		7.39 ± 0.04	7.38 ± 0.03	0.652
Mechanical ventilation	Yes	15 (39.5)	13 (34.2)	0.714
	No	4 (10.5)	6 (15.8)	
Hb (g/dL) (mean ± SD)		10.00 ± 2.01	9.14 ± 1.71	0.428

**Table 12 jcm-10-02560-t012:** Univariate Analysis of Variables from ICU Patients on Infection.

		Infection		*p*
		Yes (*n* = 15)	No (*n* = 28)	
SpO_2_ (%) (mean ± SD)		97.4 ± 3.0	98.4 ± 1.3	0.651
FiO_2_ (%) (mean ± SD)		59.5 ± 22.4	55.2 ± 21.4	0.506
pO_2_ (mmHg) (mean ± SD)		152.8 ± 66.2	152.8 ± 87.2	0.713
pCO_2_ (mmHg) (mean ± SD)		41.4 ± 7.5	39.7 ± 5.7	0.948
pH (mean ± SD)		7.394 ± 0.010	7.385 ± 0.049	0.628
Mechanical ventilation	Yes	9 (24.3)	19 (51.4)	0.703
	No	3 (8.7)	6 (16.2)	
Hb (g/dL) (mean ± SD)		10.01 ± 1.65	9.45 ± 2.15	0.500

**Table 13 jcm-10-02560-t013:** Univariate Analysis of Variables from ICU Patients on Infection with problematic Germs.

		Infection with Problematic Germs		*p*
		Yes (*n* = 4)	No (*n* = 39)	
SpO_2_ (%) (mean ± SD)		96.3 ± 4.5	98.3 ± 1.4	0.876
FiO_2_ (%) (mean ± SD)		46.7 ± 4.7	57.8 ± 22.7	0.600
pO_2_ (mmHg) (mean ± SD)		138.5 ± 1.5	154.0 ± 83.6	0.652
pCO_2_ (mmHg) (mean ± SD)		40.5 ± 0.5	40.3 ± 6.7	0.702
pH (mean ± SD)		7.400 ± 0.000	7.387 ± 0.041	0.453
Mechanical ventilation	Yes	2 (5.4)	26 (70.3)	0.578
	No	1 (2.7)	8 (21.6)	
Hb (g/dL) (mean ± SD)		No data	9.66 ± 1.99	No data

## Data Availability

The data presented in this study are available on request from the corresponding author.

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
