# Peer review of "Intensive Care as an Independent Risk Factor for Infection after Reconstruction and Augmentation with Autologous Bone Grafts in Craniomaxillofacial Surgery: A Retrospective Cohort Study"

_jcm, 2021, doi:10.3390/jcm10122560_

Round 1

Reviewer 1 Report

This is an excellent article perfectly written with an adapted methodology and an exemplary statistical analysis.
The subject matter is very interesting for specialists working in the field of facial reconstruction. I recommend the publication of this article.

 The authors of this article hypothesize that: a stay on Intensive Care Unit (ICU) might be an independent risk factor for complications, especially for infections after autologous reconstruction in OMFS (Oral and MaxilloFacial Surgery) and, the aim of this study is "to examine whether there is a significant correlation between stay on the intensive care unit and other possible risk factors such as comorbidities and underlying bony disease with the occurrence of complications after autologous mandib-ular bone grafts." The methodology used is as follows: unicentric cohort analysis: patients who required mandibular reconstruction or augmentation with autologous bone graft and fixation with ostheosynthesis devices for 2008 to 2018 from the Department of Oral and Maxillofacial Surgery at the University of Marburg . This retrospective study was approved by an ethical committee. A univariate and multivariate statistical analysis were performed.
33/128 patients exhibited graft complications needing surgical revision. Infections were most frequent. Multivariate analysis shows radiotherapy, obstructive pulmonary disease and length ofd efect as independent risk factors. Intensive care was identified as the most relevant risk factor for infection. The authors recommend best possible hygienic measures during ICU sejour.
The strengths of this manuscript are: - large number of patients
- really adapted methodology with in particular a complete univariate and multivariate statistical analysis
- the results are consistent and answer only the questions asked.
- the discussion is complete and provides the necessary comments on the results. - I honestly have no points of improvement to suggest for this manuscript. As a surgeon, this article fits perfectly into the OMFS reconstruction literature. However, I am not an expert in providing a review of the statistical tests used or in judging the quality of English because I am not a native speaker.

Author Response

Dir sir or madam,

thank you very much for your constructive criticism. We appreciate the time and effort that you dedicated to providing feedback on our manuscript and are grateful for the insightful comments.

With kind regards

The authors

Reviewer 2 Report

This study aims at investigating possible risk factors associated with the occurrence of complications to improve the outcome of autologous bone graft reconstructive procedures. Thank you for being honest about the postoperative complications. In addition, infection, radiotherapy, OPD, length of defect and ICU care etc were cited as factors that could cause complications. These factors are important and worth considering. But the most important thing isn't that happening at the time of surgery? And it will be greatly affected the individuality of each patient including medical history. This study is driving such a complex problem into a simple and unpretentious problem of only ICU care. So, will dressing and drainage changes on ICU solve all the problems?

Author Response

Dir sir or madam,

thank you very much for your constructive criticism. We appreciate the time and effort that you dedicated to providing feedback on our manuscript and are grateful for the insightful comments on and valuable improvements to our paper. Please see below for a point-by-point response to your comments and concerns. All line numbers refer to the revised manuscript file with tracked changes.

Review:
This study aims at investigating possible risk factors associated with the occurrence of complications to improve the outcome of autologous bone graft reconstructive procedures. Thank you for being honest about the postoperative complications. In addition, infection, radiotherapy, OPD, length of defect and ICU care etc were cited as factors that could cause complications. These factors are important and worth considering. But the most important thing isn't that happening at the time of surgery? And it will be greatly affected the individuality of each patient including medical history.

Author response: Thank you for pointing this out. You are correct and like you we considered it as important to collect a wide range of data from the patients. Thus, in addition to surgery related data, we also collected patient related data, which is not specifically related to the time of surgery but to the patient history in general. This should make it possible to collect a as wide base of data as possible from the patients.

This study is driving such a complex problem into a simple and unpretentious problem of only ICU care. So, will dressing and drainage changes on ICU solve all the problems?

Author response: We agree with your assessment. This is a complex problem, which we try to investigate with the broadest possible data from daily routine. It is not our intention to name the intensive care as the only constant risk factor. Multivariate analysis shows that radiotherapy, obstructive pulmonary disease and length of defect as independent risk factors. Intensive care was identified as the most relevant risk factor for infection. It is more our intention to point out the so far unknown correlation between ICU and the occurrence of complications, especially of infections besides the other known risk factors. From this we derive the clinical recommendation of optimized antisepsis in ICU as an available option in practice: hygiene rules observation, antisepsis, train and inform ICU staff and reduce duration of ICU stay as much as possible. Whether these measures reduce the occurrence of complications and infections must be investigated in the future.

Based on your comments, we have added the following paragraph to the limitations in the discussion (line 382-392): Although we did our best to assess a wide range of surgery and patient related data, an investigation of such a complex problem, due to its retrospective design, is limited by depending on data collected via daily routine work. A variety of individual factors re-lated to both surgical procedure and patients’ individual predisposition, therefore, may go unnoticed. We are aware that radiotherapy, OPD and length of defect are just exemplary drivers for the occurrence of complications and that intensive care is not the only factor responsible for the occurrence of infections. Nevertheless, these factors are results with statistical significance from the univariate and multivariate analysis and could be shown to be independent risk factors. The purpose of this article is to draw attention to this yet unknown correlation, especially regarding the correlation between infections and stay on ICU.

Reviewer 3 Report

The authors present a logical, well-constructed manuscript proposal of their retrospective analysis of risk factors for autologous craniofacial bone graft failure. The reviewer feels that with careful English language proofreading, including improved paragraph construction, this will be an imminently publishable and valuable paper. The authors are cautioned to avoid using the word prove, which is reserved exclusively for theoretical mathematics, and should never be used in empiric science. Better word choices would include: provides evidence for, suggests, findings of, data show or results indicate. The authors are also cautioned to avoid using the word assume or making any unsubstantiated assumptions where data do not clearly provide a basis for those conclusions.

Author Response

Dir sir or madam,

thank you very much for your constructive criticism and linguistic tips. We have implemented and improved them in our manuscript: line 17, 19, 270, 274, 307, 379 and 402.

We appreciate the time and effort that you dedicated to providing feedback on our manuscript and are grateful for the insightful comments on and valuable improvements to our paper.

With kind regards

The authors

Round 2

Reviewer 2 Report

The authors themselves pointed out the limitations of this paper as follows. (line 382-388)

Although we did our best to assess a wide range of surgery and patient related data, an investigation of such a complex problem, due to its retrospective design, is limited by depending on data collected via daily routine work. A variety of individual factors related to both surgical procedure and patients’ individual predisposition, therefore, may go unnoticed. We are aware that radiotherapy, OPD and length of defect are just exemplary drivers for the occurrence of complications and that intensive care is not the only factor responsible for the occurrence of infections.

There are many unnoticed variety of individual factors in this study.